# Polarization bandgaps and fluid-like elasticity in fully solid elastic metamaterials

Guancong Ma[1,2,*], Caixing Fu[1,*], Guanghao Wang[3], Philipp del Hougne[1], Johan Christensen[4], Yun Lai[3] & Ping Sheng[1,2]

Elastic waves exhibit rich polarization characteristics absent in acoustic and electromagnetic waves. By designing a solid elastic metamaterial based on three-dimensional anisotropic locally resonant units, here we experimentally demonstrate polarization bandgaps together with exotic properties such as 'fluid-like' elasticity. We construct elastic rods with unusual vibrational properties, which we denote as 'meta-rods'. By measuring the vibrational responses under flexural, longitudinal and torsional excitations, we find that each vibration mode can be selectively suppressed. In particular, we observe in a finite frequency regime that all flexural vibrations are forbidden, whereas longitudinal vibration is allowed—a unique property of fluids. In another case, the torsional vibration can be suppressed significantly. The experimental results are well interpreted by band structure analysis, as well as effective media with indefinite mass density and negative moment of inertia. Our work opens an approach to efficiently separate and control elastic waves of different polarizations in fully solid structures.

[1] Department of Physics, Hong Kong University of Science and Technology, Clear Water Bay, Kowloon, Hong Kong. [2] Institute for Advanced Study, Hong Kong University of Science and Technology, Clear Water Bay, Kowloon, Hong Kong. [3] College of Physics, Optoelectronics and Energy & Collaborative Innovation Center of Suzhou Nano Science and Technology, Soochow University, Suzhou 215006, China. [4] Instituto Gregorio Millán Barbany, Universidad Carlos III de Madrid, Avenida de la Universidad 30 Leganés, Madrid 28916, Spain. * These authors contributed equally to this work. Correspondence and requests for materials should be addressed to G.M. (email: phmgc@ust.hk) or to Y.L. (email: laiyun@suda.edu.cn).

The past decades witnessed a revolution in the study of classical waves brought about by man-made sub-wavelength composite structures, denoted metamaterials, which hold the potential of unprecedented functionalities that go far beyond those offered by Nature. The study of metamaterials has thrived in electromagnetism[1,2], optics[3,4], fluid-borne acoustics[5,6] and also in structure-borne elastic waves[7]—a ubiquitous type of classical waves that has pivotal importance in many areas of studies and applications, such as mechanical and civil engineering, geophysics and seismology. Many intriguing phenomena that were first discovered in electromagnetism, such as cloaking[8–10], negative refraction[11,12], super-resolution focusing[12,13], have been successfully demonstrated in thin plates. 'Pentamode' metamaterials demonstrate extreme mechanical properties through intricate designs of strut frameworks[14–21]. Ambitious projects also aim to reduce the destructive power of seismic waves[22–24]. However, elastic waves are distinct from both electromagnetic and acoustic waves by possessing a richer variety of polarization characteristics—waves with both longitudinal and transverse nature are allowed[25]. While these characteristics bring new features, they also make elastic waves more complex and more difficult to handle than electromagnetic and acoustic waves.

Sub-wavelength building blocks with local resonances[26,27] have been widely utilized for the design of acoustic metamaterials. Double negativity in acoustics[28–30], sub-wavelength wave guiding[31], super-lensing[32,33] and super-absorption[34,35], are examples among a plethora of exotic functionalities. Remarkably, recent theoretical studies linking local resonance with elastic waves in two-dimensional systems have found fascinating consequences and unique properties, such as negative shear modulus[36], fluid-like behaviour[37], super-anisotropy[37] and so on. However, because of the structural complexity, these fascinating systems present tremendous experimental challenges. Consequently, the appeal of resonant elastic metamaterials with unique properties and richer physics in polarization control beyond their acoustic and electromagnetic counterparts mostly remained theoretical.

Here we present the design and experimental realization of a type of three-dimensional locally resonant elastic metamaterial, which exhibits polarization bandgaps and 'fluid-like' elasticity. Based on such metamaterials, we construct unique rod-shape structures, which we denote as 'meta-rods'. By measuring their response functions for flexural, longitudinal and torsional vibrations, we demonstrate the selective suppression of these vibrations by polarization bandgaps and unprecedented elastic rod properties. In particular, we observe that in a certain frequency regime, only the longitudinal vibration can be excited in the meta-rod, whereas flexural vibrations are forbidden—a hallmark elastic property of fluids (hence denoted 'fluid-like' elasticity). Whereas in another configuration of the same metamaterial units, the meta-rod can significantly suppress the torsional vibration in a certain frequency regime. These unusual characteristics revealed by experiments are well interpreted by band structures as well as effective medium theory that exhibits negative moment of inertia and indefinite mass densities.

## Results

**Metamaterial design.** A photographic image of the unit cell is shown in Fig. 1a. A steel cylinder, with a radius $r = 15.8$ mm and height $h = 37.6$ mm, is coated with silicone rubber. The cylinder's axial direction is defined as the $z$ axis, and the $xy$-plane is parallel to the end surfaces. The silicone layers on the top and at the bottom of the steel cylinder are both 1 mm in thickness; whereas the silicone covering the curvilinear side has a thickness of 5 mm. The silicone-coated steel cylinder is then cast inside an epoxy cube with each side $a = 60$ mm. The physics of this unit cell can be understood with a simple mass-spring-mass model, as shown in the inset of Fig. 1a. Here the steel cylinder and the epoxy cladding mostly have the role of two block masses, $M_1$ and $M_2$, which interact through compression and expansion of the silicone rubber

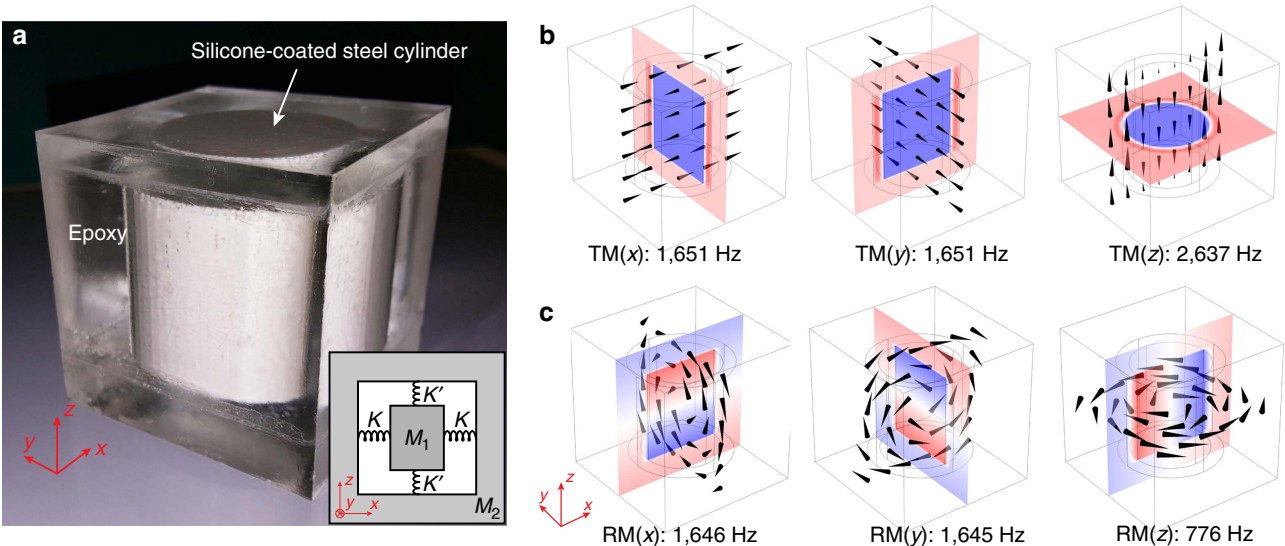

**Figure 1 | The unit cell and its eigenmodes.** A photographic image of the unit cell is shown in **a**. A simple spring-mass model of the system is shown in the inset, wherein $M_1$, $M_2$ are the mass of the core (steel cylinder), and shell (epoxy), respectively; $K$ and $K'$ are different spring constants, which are tunable via the thickness of the silicone coating at relevant positions. Simulated displacement profiles of the three translational modes (TMs) are shown in **b**. Two modes are degenerate at 1,651 Hz, denoted by TM($x$) and TM($y$), in which the steel and epoxy oscillate anti-phase in the $xy$-plane. A third mode, denoted TM($z$), is found at 2,637 Hz, in which the steel core and epoxy oscillate anti-phase in the $z$-direction. Three rotational modes (RMs) are shown in **c**, characterized by the out-of-phase rotational oscillation of the steel and epoxy along a certain axis (indicated in the parentheses). In **b,c**, black cones indicate the amplitude and direction of the displacement. Colours represent the displacement component perpendicular to the slicing plane, with red/blue representing positive/negative displacement, respectively. All eigenmodes were calculated using an isolated unit cell.

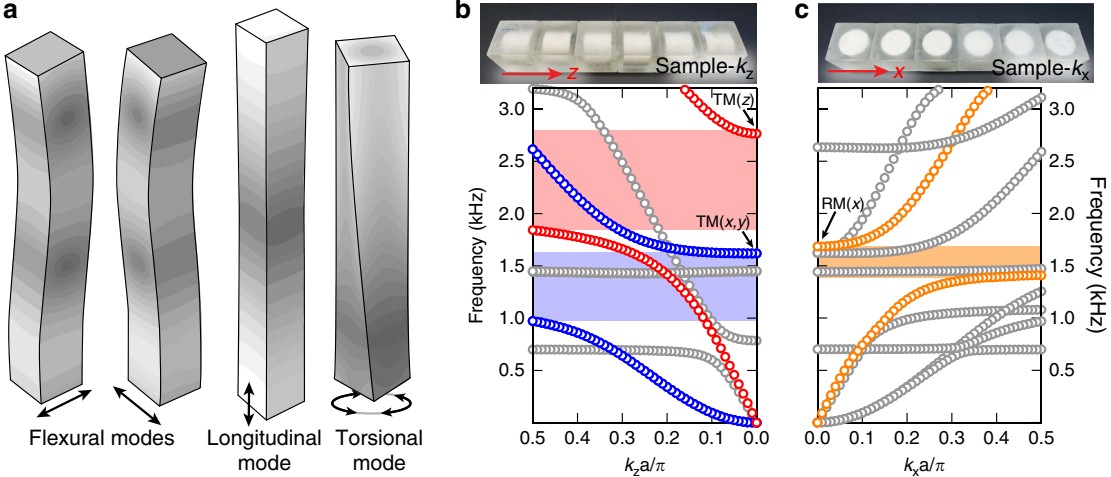

**Figure 2 | Vibration modes of a rod and the band structures of meta-rods.** (**a**) Shows schematic drawings of four fundamental vibration modes of an ordinary rod. Two are flexural modes, characterized by the bending of the rod under excitation forces that are transverse to the rod. One is a longitudinal mode, in which the particles dominantly oscillate along the rod, accompanied by a small degree of breathing. Torsional mode describes the rod's twisting oscillation along itself. Two different types of meta-rods are shown in **b**, **c** (top), together with their band structures (bottom). Here blue/red markers highlight the flexural/longitudinal branches in sample-$k_z$. Blue/red shaded regions are flexural/longitudinal bandgaps, respectively. In **c**, orange markers highlight the torsional branch of sample-$k_x$. A torsional bandgap is shaded in orange.

that acts as springs. Owing to the difference in the silicone's thickness, the spring constants in relevant positions are different (denoted by $K$ for the side, and $K'$ for top and bottom, respectively).

The consequence of the difference in silicone thicknesses or spring constants can be immediately seen by investigating the eigenmodes of an isolated unit cell using finite-element simulations. Three translational modes (TMs) are found, in which the steel cylinder undergoes translational vibration and moves out-of-phase with respect to the epoxy cladding, which are shown in Fig. 1b. Two of these modes are degenerate at 1,651 Hz, in which the cylinder and the epoxy vibrate in the $xy$-plane. We denote these two modes as TM($x$), TM($y$). The third mode, wherein the steel cylinder and the epoxy vibrate in the $z$-direction, has a much higher eigenfrequency at 2,637 Hz. This mode is denoted by TM($z$). The large mismatch in eigenfrequencies between TM($z$) and TM($x, y$) is mainly due to the difference in the silicone coating's thickness. Basically, a larger thickness means a smaller spring constant, thereby leading to a lower eigenfrequency; whereas a smaller thickness indicates the opposite. By changing the thickness of the silicone rubber, we can conveniently engineer the corresponding eigenfrequencies independently. This is shown in Supplementary Fig. 1 and Supplementary Note 1.

Three rotational modes (RMs) are also found in simulations, wherein the steel cylinder and epoxy oscillate rotationally about each of the three spatial axes in an anti-phase manner, as shown in Fig. 1c. Their physical consequences are demonstrated and discussed below.

**Quasi-one-dimensional arrays as meta-rods**. By assembling the unit cells into a quasi-one-dimensional periodic array, we obtain an elastic structure commonly known as a rod. An elastic rod can sustain four distinct branches of waves: two flexural, one longitudinal and one torsional[25,38]. They are easily distinguishable through their vibration profiles, as schematically illustrated in Fig. 2a. Characterized by the rod's bending, flexural modes are dominantly transverse in their nature, therefore, there are two orthogonal polarizations. In the longitudinal vibration, displacements in the rod mainly are along the axial direction,

accompanied by a small amount of breathing. In the torsional mode, a rod engages in rotational and twisting vibration along its own axis. These three types of vibrations can be identified through their dispersion signatures at low frequencies. The longitudinal and torsional waves have linear dispersions. Flexural waves, however, are governed by biharmonic equations[25,38], therefore, have distinctive quadratic dispersion relations. As we shall next demonstrate, the metamaterial unit cells endow the rods with very unusual vibration properties, hence they are denoted as 'meta-rods'.

From the unit cell's geometry, it is straightforward to see that there exist two different ways to build the quasi-one-dimensional array. One is to repeat the unit cell in the $z$-direction, that is, along the steel cylinder's axis. We denote this array as 'sample-$k_z$'. The second type of array, denoted by 'sample-$k_x$', has periodicity in the $x$ (or $y$) direction, that is, perpendicular to the steel cylinder's axis. These samples are shown in Fig. 2b,c (top). Next, we demonstrate the emergence of unique vibrational properties such as the intriguing fluid-like elasticity, and the extinction of torsional vibration.

**Fluid-like characteristic**. First, we excite sample-$k_z$ by a force perpendicular to the rod. This is realized by the experimental set-up shown in Fig. 3a. Simply put, the sample is uprightly fixated on an aluminum plate that is supported on low-friction sliding tracks, which confine their motion to one single direction. We then connect the aluminum plate to an electromagnetic shaker, whose vibration causes the plate to move back and forth along the tracks (see Methods for more details). For the meta-rod, this transverse excitation will excite flexural vibrations. We obtain the response function by dividing the acceleration (with direction along the excitation force) measured on the top of the sample with the one measured on the aluminum plate. This is plotted in Fig. 4a as a function of frequency (blue circles). A bandgap is clearly seen in ∼1.2–1.6 kHz. We further use a laser Doppler vibrometer to map the displacement parallel to the actuation direction on a facet of the meta-rod at 1,350 Hz, that is, inside the bandgap. The result is shown in Fig. 4b. It is seen that the displacement amplitude rapidly decays away from the actuation position ($z = 0$ mm). This means that unlike ordinary elastic rods,

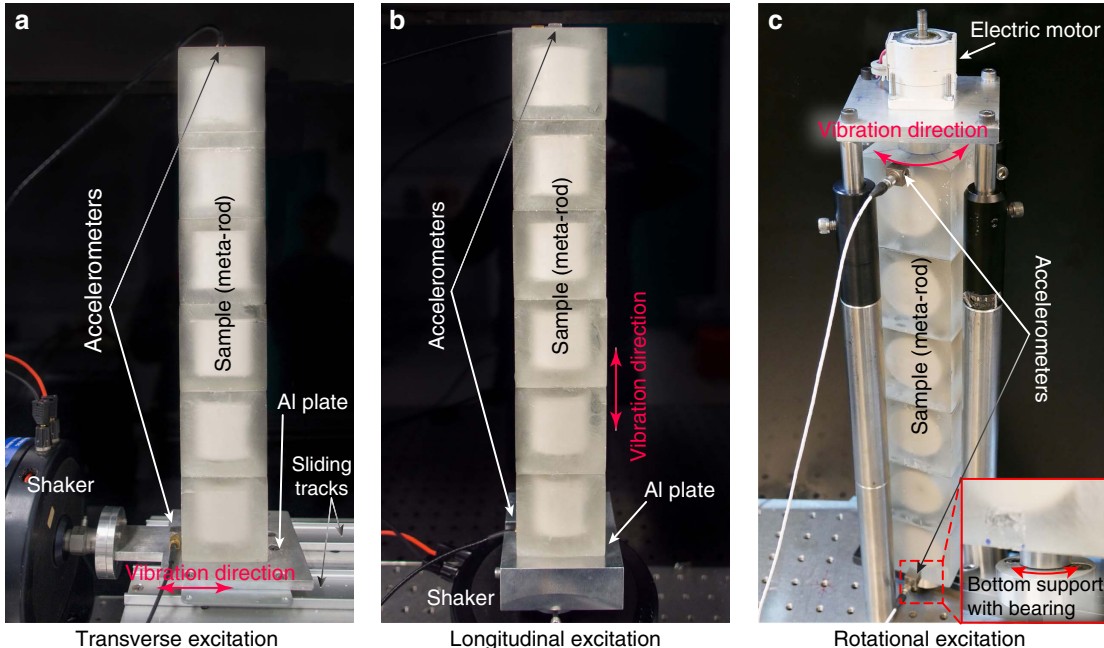

**Figure 3 | Experimental set-ups.** An electromagnetic shaker is used for transverse excitation in **a**; and longitudinal excitation in **b**. Rotational excitation is achieved by using an electric motor, which exerts a torque pulse about the meta-rod's axis (x axis), as shown in **c**. Accelerometers are used to obtain the elastic responses at different positions.

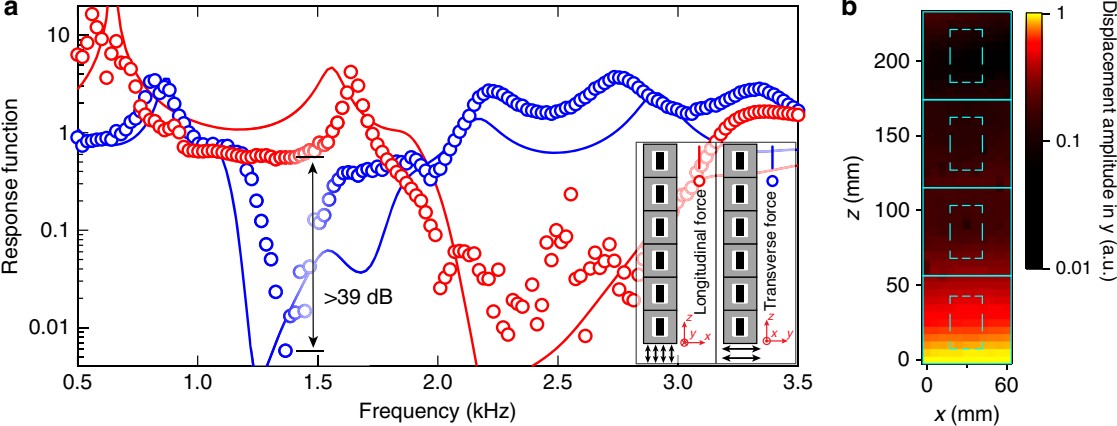

**Figure 4 | Fluid-like elastic characteristic of the meta-rod with periodicity in z.** (**a**) Shows measured response functions. Red/blue represents longitudinal/flexural response, respectively; makers/curves represent measured/simulated results. Spectrally mismatched bandgaps are clearly observed. Specifically, a bandgap for flexural waves is seen in ~1.2–1.6 kHz, which overlaps with longitudinal waves passband. The response difference exceeds 39 dB (~1,350 Hz). This means only longitudinal waves can be sustained in this frequency regime, an intriguing elastic property like fluids. Slight mismatches between experiment and simulations are attributed to fabrication errors. (**b**) Shows the out-of-plane displacement field of sample-$k_z$ under transverse excitation. The frequency is 1,350 Hz, that is, inside the flexural bandgap. The colour map represents normalized displacement amplitude. The cyan solid/dashed lines delineate the positions of the unit cells/steel cylinders, respectively. The aluminum plate for excitation is situated at z = 0 mm.

transverse forces with frequencies inside the bandgap cannot excite the flexural vibration in our sample.

Next, the same sample is subjected to a pushing and pulling force exerted along the rod's axis. To achieve this, the meta-rod is fixed on a thick aluminum plate, which is connected to the shaker, as shown in Fig. 3b. The shaker excites the longitudinal branch in the meta-rod. The measured response function is shown with red circles in Fig. 4a. A bandgap is found covering ~1.8–2.9 kHz, which is spectrally mismatched with the flexural bandgap. Comparing the sample's flexural and longitudinal responses, an intriguing spectral regime can be found in ~1.2–1.6 kHz. In this regime, the meta-rod can only withstand the longitudinal vibration. Transverse vibrations cannot

be sustained. The response difference between these two polarizations exceeds 39 dB around 1.4 kHz. Traditionally, such a characteristic is expected only in fluids, and is not found in any solid material. We have also carried out numerical simulations for both excitations. The results are shown in Fig. 4a. Good agreement with experiments is seen. Slight mismatches are attributed to fabrication errors.

The underlying physics of this fluid-like elasticity can be understood by the band structure, as shown in Fig. 2b. A force transverse to the meta-rod excites a flexural branch, which can be identified via its quadratic dispersion at the low-frequency limit. In sample-$k_z$, the transverse force will also excite one of the TM(x, y), in which the motion of steel core/epoxy are in the

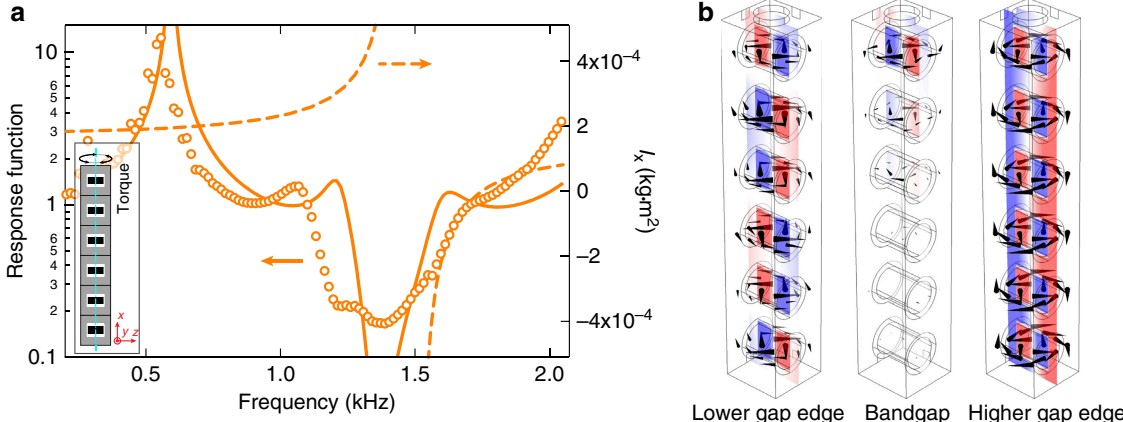

**Figure 5 | Torsional response and negative effective moment of inertia.** The response function is defined as the ratio of the amplitudes of tangential accelerations at two ends of the sample, with rotational actuation situated at the top the sample (Fig. 3c). The measured (orange markers) and simulated response functions (orange solid curves) are plotted in **a** as functions of frequency (left axis). A bandgap is seen near 1.3–1.6 kHz. The calculated effective moment of inertia $I_x$ is shown in orange dashed curves (right axis). It is seen that $I_x$ turns negative inside the bandgap. Inset shows a drawing of the sample-$k_x$. (**b**) Shows the simulated displacement profile of the sample-$k_x$ at the lower gap edge (1.3 kHz), inside the bandgap (1.5 kHz), and at the higher gap edge (1.7 kHz). Here actuation position is on the top of the meta-rod. Black cones indicate the amplitude and direction of the displacement (in logarithmic scales). Colour fields represent the displacement component that is perpendicular to the slicing plane, with red/blue colours representing positive/negative displacement, respectively.

$xy$-plane. This mode couples with flexural vibrations. Their anti-crossing produces a polariton-like dispersion (blue markers, Fig. 2b) and a bandgap (shaded blue in Fig. 2b.) Similarly, a force along sample-$k_z$ excites the longitudinal branch together with TM($z$) (red markers, Fig. 2b). Owing to the thinner silicone rubber at the two ends of the steel cylinder, the longitudinal bandgap opens at much higher frequencies (shaded red in Fig. 2b). The flexural bandgap, therefore, overlaps with the longitudinal pass band, giving rise to a region that only sustains longitudinal vibrations. Owing to the isotropy in sample-$k_z$ in the $xy$-plane, this characteristic disregards the polarization of flexural waves. Fluid-like elasticity is the consequence.

By simply changing the orientation of the same unit cells, sample-$k_x$ has distinctive features. Detailed experimental results of numerical investigations and relevant discussions can be found in the Supplementary Fig. 2 and Supplementary Note 2. In short, a 'partially fluid-like' region is observed in the frequency range covering ∼1.7–2.5 kHz. In sample-$k_x$, the longitudinal branch couples with TM($x$); and flexural branches interact with TM($z$) or TM($y$), depending on the polarization. Also, the eigenfrequencies of TM($x$) and TM($y$) are almost identical (they are degenerate in an isolated cell), therefore, fluid-like elasticity only exists for the $z$-polarized flexural waves. In practice, this characteristic can potentially be utilized to switch from fluid-like to solid-like—simply rotating sample-$k_x$ along its axis ($x$ axis) by 90°.

**Torsional bandgap and negative moment of inertia.** RMs are found in our unit cell (Fig. 1c). RMs have long been theoretically studied in phononic crystals and metamaterials[37,39–43]. Yet, owing to the difficulty in excitation, direct observations of these unique modes and their consequences are still lacking. The meta-rod's geometry offers an interesting opportunity to observe and to appreciate the physical significance of RMs via torsional vibration (Fig. 2a).

We choose sample-$k_x$ to investigate the effect of RM on torsional vibration. First, the meta-rod's torsional branch can be excited by a dynamic torque about the $x$ axis. The same torque can also trigger RM($x$), in which the steel cylinder and epoxy rotate about the $x$ axis in an anti-phase manner. RM($x$) is found at 1,646 Hz (Fig. 1c). Similarly, the anti-crossing due to counter-rotating modes (RM($x$) and the torsional branch) yields a paritonic dispersion. Through the torsional branch's linear dispersion signature and the system's rotational displacement profile, this can be identified in the band structure (orange markers, right panel of Fig. 2c). A bandgap for torsional vibration is also highlighted in orange in Fig. 2c. Experimentally, we verify this torsional bandgap by rotational actuation with a set-up shown in Fig. 3c, in which an electric motor is used to apply a torque pulse to the meta-rod. The measured response function indeed confirms the existence of such a bandgap, as shown in Fig. 5a. The measured results show good agreement with numerical simulation.

We can gain more insights about the meta-rod's torsional behaviours from its displacement profiles at certain key frequencies, which are plotted in Fig. 5b. At the lower gap edge, the meta-rod undergoes pronounced twisting. However, in each unit cell, the steel cylinder vibrates in phase with respect to its epoxy cladding. Inside the bandgap, torsional vibration exponentially decays away from the actuation position. At the high gap edge, we can clearly observe that all steel cylinders are rotating out-of-phase with respect to the epoxy. These observations resemble closely the mode profile of acoustic/elastic metamaterials with negative effective mass density[6]. In a similar spirit, effective moment of inertia of the meta-rod per unit, $\overleftrightarrow{I}$, can be defined as $\Delta\vec{\tau} = \overleftrightarrow{I} \cdot \vec{\alpha}$. Here $\Delta\vec{\tau}$ is the effective torque applied on the unit, and $\vec{\alpha}$ is the effective angular acceleration of the unit. By using the eigenfunction of RM($x$), we can easily calculate the $x$ component of inertia $I_x$. The result is plotted in Fig. 5a as a function of frequency. It can be seen that $I_x$ is negative near 1.5 kHz. For torsional vibration, relevant wave parameters are the moment of inertia and torsional rigidity. Torsional waves exponentially decay if the rod has single negativity in moment of inertia. This is in alignment with both observation and analysis. The extraction scheme of effective moment of inertia is shown in Supplementary Note 3.

A torsional bandgap is also numerically found in sample-$k_z$ (Supplementary Fig. 3), wherein RM($z$) has an important role. However, we did not observe this bandgap experimentally. We speculate that the discrepancy probably owes to the non-ideal bonding between the steel/silicone/epoxy interface(s) around the

curvilinear surface, which may result in slipping in circumstances of torsional vibration, and, therefore, leads to ineffectiveness of excitation. Relevant discussions are included in Supplementary Note 4.

## Discussion

The elastic properties of our meta-rods are vastly different from rods made of traditional solid materials. Yet they all arise through a simple mechanism: the anti-crossing between local resonances in metamaterial units and a certain vibration branch of the rod. Two types of eigenmodes are considered: translational and rotational. For TMs, the structure is in fact quite similar to the design first reported by Liu *et al.*[26], that is, a spherical metal core evenly coated with a layer of silicone rubber, and embedded in an epoxy cube. Clearly, the unit cell in ref. 26 has a high-degree of symmetry, and is isotropic in all three spatial dimensions. Although a low-frequency bandgap can be created due to negativity of mass density, the selection of vibration modes cannot be realized. Here by breaking spatial symmetry, the metamaterial becomes anisotropic. The consequence is that the bandgaps for different polarizations are spectrally mismatched, which manifest fluid-like behaviour in a certain frequency regime, despite that all constituents are solid. This unique phenomenon is a direct consequence of the richer freedoms of polarizations existing in elastic metamaterials than that in acoustic and electromagnetic metamaterials. Moreover, because this characteristic is achieved via symmetry breaking, it is robust against perturbations in material properties as well as flaws in fabrication. These factors translate into the ease of implementation. We further demonstrate with numerical studies the robustness of these functionalities in systems with different geometric shapes (Supplementary Figs 6,7 and Supplementary Note 6), and different materials (Supplementary Figs 8,9 and Supplementary Note 7). These results show that the polarization bandgaps and the related functionalities can be easily attained with a large variety of material choice and geometry configurations. Such an unusual property expands the horizon of elastic wave manipulations. For example, it may improve impedance matching between solids and fluids as well as soft tissues, in which transverse or shear waves cannot propagate. It may also be exploited as a polarizer for elastic waves.

We note that similar fluid-like behaviours were theoretically shown to arise in two-dimensional systems via intricate hybridization of multipole local resonances[37]. However, the microstructure is highly complex, therefore, presents a tremendous fabrication challenge. The approach adopted here is much more convenient and practical. Dirac-cone dispersion in phononic crystals was also theoretically shown capable of similar characteristics[44]. Nevertheless, the system relies on a zero refractive index as well as accidental degeneracy, which imposes high requirement of system parameters, making it less robust against perturbations, and, therefore, is also difficult to realize. In addition, lattice constants in phononic crystals are at the same order of wavelength, whereas the lattice constant in our design is deep sub-wavelength. On the other hand, 'pentamode' metamaterials, essentially skeletons of elongated struts joint together at their tips so as to maintain rigidity, also demonstrated vastly different transverse and compressional elastic modulus[14,16]. These delicate structures are not fully solid, and require advanced fabrication techniques[17–19]. Existing experimental investigations focused more on their static mechanical properties[15–18,45] and shock responses[21].

The metamaterial units can also be assembled into a three-dimensional periodic lattice. It is seen that the properties associated with TMs also emerge: spectrally mismatched polarization bandgaps for longitudinal and shear waves are found, and fluid-like elasticity exists. However, the effects of RMs mostly disappear, since torsional branches do not exist in a bulk solid. In a three-dimensional lattice, the fluid-like property can be perfectly accounted for by effective medium. Here TMs have dipolar symmetry hence they introduce anomalous effective mass densities[6,37,46,47]. Owing to the mismatched eigenfrequencies of TM(z) and TM(x, y), the effective mass density tensor is extremely anisotropic: we found that certain component(s) of the mass density tensor can become negative while other components remain positive. Similar to indefinite permittivity in electromagnetic systems[48], these 'indefinite effective mass densities' directly associate with the elastic wave propagation. In fluid-borne acoustics, such indefinite mass density usually can be realized by layer structures and lead to hyperbolic dispersions[49–52], that is, sound cannot propagate in the direction wherein the effective mass is negative. Likewise, elastic waves cannot propagate a medium with single-negative effective mass[12,13]. However, in the context of elastic waves, the physical consequences of this indefinite mass density is richer, and can be used to attain polarization control. Numerical results of a three-dimensional array and detailed discussions are presented in the Supplementary Figs 4,5 and Supplementary Note 5.

The meta-rod has another particularly interesting aspect in its ability to resist shaking that is transverse to its axis, that is, flexural vibration, at a deep sub-wavelength scale. In the particular geometry of a rod, our design may hold great potential in civil engineering. Flexural vibration is a major source of potential damage for manmade structures during earthquakes. In the experiment demonstrated in Fig. 3a, we emphasize that the long thin meta-rod was fixed only at the bottom to a plate that shakes horizontally—a configuration that faithfully reflects many free-standing structures, such as pillars, skyscrapers, towers, chimney, and so on. Yet this meta-rod shows resilience to lateral shaking within the frequency regime of the flexural bandgap. It can be seen in Fig. 4b that the vibration decays within one unit cell, whose dimension is at the deep sub-wavelength scale (wavelength of flexural wave at $\sim 1.3\,$kHz in epoxy exceeds one meter). Therefore, counter-intuitively, the parts of the meta-rod that are further away from the ground ($z = 0$) are more stable. The vibration shielding functionality of meta-rods has been videotaped and is demonstrated in Supplementary Movie 1. In comparison, existing seismic protective solutions employ sophisticated systems such as the massive roof-loaded 'seismic damper' to protect skyscrapers. 'Seismic shields' installed in the earth[22,24] will have to mitigate both surface waves and body waves, and have to deal with complexities such as the soil and rock properties[53,54]. Our results suggest elastic metamaterials/meta-rods as an interesting alternative. A similar vibration shielding effect can also be obtained for longitudinal vibrations (a demonstration in Supplementary Movie 2). We note that the condition of anisotropy is not necessary for vibration shielding applications. Using isotropic unit cells can bring longitudinal and flexural bandgaps to overlap, creating a full bandgap capable of stopping vibrations of all polarization types.

On the other hand, RMs and the negative effective moment of inertia provide intriguing possibilities to mitigate torsional vibrations, which have been recognized as a potential hazard in machineries with rotating parts, such as drills, propulsion shafts, and so on.

## Methods

**Sample fabrication.** The metamaterials were fabricated one unit cell at a time. Unit cells were joined together by acrylic super-glue. The materials are: Wacker silicone rubber RTV-2 Elastosil M4440, Clear Casting Floral Arrangements decorative epoxy resin.

**Measurements.** For the measurement of the response spectra under transverse and longitudinal excitations, we used a waveform generator (Agilent 33220A) to send a pulse covering 0.5–3.5 kHz to an electromagnetic shaker (Brüel and Kjær Vibration Exciter type-4809) through a power amplifier. The samples were super-glued onto an aluminum plate. For transverse excitation, the sample and the aluminum plate were supported by two parallel low-friction sliding tracks, which confined the motion to one single direction. A facet of the aluminum plate was then firmly attached to the shaker. For longitudinal excitation, a thicker aluminum plate was used so that the plate's bending modes located at frequencies higher than our range of interest. The center of the plate was directly connected to the shaker. Two tri-axial accelerometers (Dytran 3023A) were adhesive-bonded to relevant positions. Details of experimental set-ups can be more easily seen in Fig. 3a,b. The accelerometers measured the vibration along the excitation directions. The exterior of our samples are relatively rigid and deformation is minimal. Still, to obtain best results, we measured 24 points on the top surface of the sample, and used the arithmetic mean to calculate the response functions. The signals were recorded by a digital oscilloscope (Agilent DSO6014A).

To measure the torsional response, the meta-rod was fixed on a rotational stage at its axial center at the bottom. The top was fixed to a DC motor that was connected to the waveform generator. The two accelerometers measured the tangential accelerations at the top and the bottom of the sample.

The vibrational profile (Fig. 4b) was acquired as follows. A piece of aluminum kitchen foil (thickness $\sim 0.02$ mm) was adhered to cover the facet of the sample to be measured, which increased the reflectivity. A laser Doppler vibrometer (Graphtec AT500-05) was mounted on a two-dimensional motorized translation stage to scan the surface point by point. Our translational stage has limited travel range that only covers four unit cells. The laser beam was perpendicular to the measured surface, and was parallel to the excitation force. The sample was excited by the shaker driven by a sinusoidal signal, and the measured data was recorded by a lock-in amplifier (Stanford Research SR830).

**Simulations.** Numerical simulations were performed using the three-dimensional solid mechanics module in COMSOL Multiphysics (v4.3a). The band structure and eigenmodes were solved using eigenfrequency study, whereas the frequency response functions were solved by frequency-domain study. Parameters used in simulations are: for the epoxy, the Young's modulus $E = 3.8$ GPa, Poisson ratio $v = 0.350$, mass density $\rho = 1,130$ kg m$^{-3}$; for the steel cylinders, $E = 180$ GPa, $v = 0.250$, and $\rho = 7,850$ kg m$^{-3}$; for the silicone rubber, $E = 3.3$ MPa, $v = 0.477$, and $\rho = 1,245$ kg m$^{-3}$. In the calculation of frequency response functions, dissipation is added to Young's modulus of silicone rubber, that is, $E = 3.3 + 0.3i$ MPa. For the purpose of clear viewing, the displacement profiles shown in Fig. 5b were obtained without dissipation.

**Data availability.** The data in this study are available from the corresponding authors on request.

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

## Acknowledgements

G.M. and C.F. thank Zhiyu Yang for equipment and lab space. G.M. thanks C.T. Chan and Zhao-Qing Zhang for helpful discussions. G.M., C.F., and P.S. acknowledge the support of the Hong Kong Research Grants Council (Grant No. AoE/P-02/12). Y.L. and G.W. thank the State Key Program for Basic Research of China (No. 2014CB360505, No. 2012CB921501), National Natural Science Foundation of China (No. 11374224, No. 61671314), and a Project Funded by the Priority Academic Program Development of Jiangsu Higher Education Institutions (PAPD). J.C. acknowledges the support from the European Research Council (ERC) through the Starting Grant 714577 PHONOMETA.

## Author contributions

G.M. and Y.L. supervised the project. Y.L., J.C., and G.M. conceived the metamaterial design. G.M. and C.F. fabricated the samples and carried out experimental measurements. P.d.H. assisted with experiments. G.M. and G.W. performed numerical simulations. All authors were involved in analysis and discussion of the results. G.M. and Y.L. wrote the manuscript. J.C. and P.S. revised and improved the manuscript.

## Additional information

**Competing financial interests:** The authors declare no competing financial interests.

