## [Peer Review File · Nature Communications]

Reviewers' comments:

Reviewer #1 (Remarks to the Author):

The authors report frequency response of a composite comprised of an array of steel cylinder-epoxy system. By measuring the vibrational responses under flexural, longitudinal, and torsional excitations, the authors find that each vibration mode of the meta-rods can be selectively suppressed. In particular, the authors observe that in a particular frequency regime, only the longitudinal vibration can be excited in the meta-rod, whereas flexural vibrations are forbidden – a hallmark elastic property of fluids (hence denoted “fluid-like” elasticity). The work is centered on simulations and experimental observations of the three types of excitation. Although the results are intriguing, I find the work lacking of breadth which makes these effects limited to particular types of material constituents and specific, selected vibration excitation as indicated in the paper. The following questions need to be addressed before a recommendation can be considered.

What is the effect of the aspect ratio of the rod on the band gap properties of the “meta-rod”? What is the effect of cross sectional geometry? (eg. square vs. circle? vs. rectangle)

What is the effect in terms of material choice and weight efficiency? i.e., how will the property change as a result of different material combinations other than the specific material pair (steel, epoxy and silicone) the authors listed?

The authors claim that these “meta-rods” can be served as an alternative method to be used for seismic shielding and demonstrated the shielding effect flexural vibrations. Can the authors comment on other types of vibrations or combinations of different vibrations?

Reviewer #3 (Remarks to the Author):

This manuscript describes the modeling and experiments associated with an “elastic metamaterial.” This consists of “meta-rods” where there are unit cells of metal cylinders encased in a polymeric matrix where there are different thicknesses of polymer on the faces of the cylinder as opposed to the curved surface. When these unit cells are arranged into rod-like structures, interesting vibrational behavior can be observed. Specifically, depending on how you orient the cylinders, the authors have been able to show band gaps in certain vibrational modes and even “fluid-like” behavior where longitudinal vibrations are transmitted while translational modes are damped. I found this paper to be interesting, novel, and very well written. The concept is well explained and the results clearly show the concept through actual testing and the discussion is easily understandable. I commend the authors on this work and a well written manuscript. This is a very interesting and powerful concept which can be applied in many ways. The authors highlight design for seismic activity at the end of the paper but there are clearly many other applications as well. This is impactful work which should be published in Nature Communications. I think that the manuscript is very complete and nearly publishable in its current state. I have the below additional comments/thoughts which are not required for

- One way to make this already excellent paper stronger would be if the authors could have drawn some more general conclusion. There must be some non-dimensional numbers and/or ratio of K/K' which define some of the behavior of the system. These generalizations could serve as a design guide to others.
- More thought/discussion on how to extend this concept to 3D would be very interesting.

- Could this concept be miniaturized and become elements within a “material”? Could the material have band gaps for specific frequencies along certain axes and provide vibrational isolation?
- Finally, while mentioning other literature, the authors state on page 12 that “...existing experimental investigations focused more on their static mechanical properties, while testing of their dynamic/wave properties are lacking.” This is still generally true but starting to change. An initial foray into this area is shown in “Dynamic Behavior of Engineered Lattice Materials,” Hawreliak, et al., Scientific Reports, 20 June, 2016.

REVIEWERS' COMMENTS:

Reviewer #1 (Remarks to the Author):

The authors have thoroughly addressed my concerns raised in the previous round of review. By adding additional data on geometric and material effect on the proposed meta-rod, this paper has substantially gained in depth and quality. I would recommend its publication in its current form.

Reviewer #3 (Remarks to the Author):

I have no additional comments and feel that the authors address everything that I mentioned in the review. I recommend the paper for publication.

Reviewer #1 (Remarks to the Author):

The authors report frequency response of a composite comprised of an array of steel cylinder-epoxy system. By measuring the vibrational responses under flexural, longitudinal, and torsional excitations, the authors find that each vibration mode of the meta-rods can be selectively suppressed. In particular, the authors observe that in a particular frequency regime, only the longitudinal vibration can be excited in the meta-rod, whereas flexural vibrations are forbidden – a hallmark elastic property of fluids (hence denoted “fluid-like” elasticity). The work is centered on simulations and experimental observations of the three types of excitation. Although the results are intriguing, I find the work lacking of breadth which makes these effects limited to particular types of material constituents and specific, selected vibration excitation as indicated in the paper. The following questions need to be addressed before a recommendation can be considered.

We are glad that the reviewer found our results intriguing. We are also thankful that the reviewer pointed out certain short-comings in our paper. Indeed, in our manuscript we focused on describing the realization and physical origin of the polarization bandgaps, and the two unique consequences in the context of elastic waves, i. e., the fluid-like elasticity and torsional bandgaps. In order to experimentally demonstrate these new concepts and their unique characteristics, we have to choose a specific example (cubic unit cell made from steel core, silicone coating and epoxy background). But certainly the idea and core results demonstrated in this paper can be applicable to more general scenarios, with different choice of materials and vibration types. In order to support these arguments, **we added two more sections to the *Supplementary Information (S-V. Geometry of meta-rod’s cross section, and S-VII. Material choice)* to include new numerical results, and relevant discussions on these matters. The main text is also slightly revised accordingly. The revised parts are highlighted.** We sincerely hope the revision can sufficiently answer the reviewer’s question, and can resolve his/her concerns.

What is the effect of the aspect ratio of the rod on the band gap properties of the “meta-rod”?
What is the effect of cross sectional geometry? (eg. square vs. circle? vs. rectangle)

This is indeed an interesting question. The consequence of changing the cross-sectional shape of the

meta-rod is many-fold. First, it would induce some change in the mass of the epoxy cladding, which will affect the eigenfrequency of each mode. Since the unit cell can be described with a mass-spring-mass harmonic resonator (inset, Fig. 1a), the eigenfrequencies approximately follow the rule $\omega \sim \sqrt{1/m}$. However, the change in the cross-sectional shape affects all relevant eigenmodes, and therefore does not present an issue to the general behaviors of the polarization bandgaps. Second, **flexural modulus and torsional rigidity of a rod are both associated with the rod's cross-sectional shape**¹. Consequently, we expect a certain degree of changes in the wave speed when a different shape is considered. This effect is manifest as a change in the slope of the flexural (blue, Fig. 2b) and torsional (orange, Fig. 2b) propagative branches in the band structure (particularly near low-frequency limit). Third, **if the shape is changed into a rectangle, then the rod becomes anisotropic in its cross sectional directions**. The consequence is that the two flexural modes (with orthogonal polarizations) must have different wave speed. This is schematically illustrated in Fig. R1.

Figure R1 For a rod with rectangular cross section, the two flexural modes are no longer degenerate and have different dispersions. If the rod the bending along the thicker direction (blue inset), the flexural modulus in effect is larger, thereby the wave speed is generally higher (blue curve). If the rod the bending along the thinner direction (red, inset), a smaller flexural modulus leads to lower wave speed (red curve). This will lead to interesting consequences for meta-rods as well.

To verify these analyses, we have performed numerical study on two meta-rod models, one with circular cross section, and the other with rectangular cross section. Here, meta-rods with circular cross section is built with 6 cylindrical unit cells, each with a diameter of 60 mm and height of 60 mm. Same as in the main text, the axis of the steel cylindrical core is defined as the z -axis. The band structure of both type of unit cell stacking (along z and x) are plotted in Fig R2a, b. (Here, different unit cells are needed for the two stacking types. In comparison, for cubic unit cells, changing stacking direction

merely means to change the unit cell's orientation.) Meta-rods with cross section of a rectangle also consist of 6 unit cells, the rectangle has a long side $b = 120\text{mm}$, and a short side $a = 60\text{mm}$ (therefore an aspect ratio of 2:1). Their band structures are plotted in Fig. R2c, d. Note that now there exist two possible cases for x -stack meta-rod: the cylindrical steel core can lie parallel to the short side (x -stack 1), or parallel to the long side (x -stack 2) of the rectangular cross section.

For the circular meta-rods, the overall behaviors are almost identical to the square meta-rods, beside some shifts in frequencies. Fluid-like region is found for z -stack meta-rod near 1.0–1.8 kHz (grey shaded region in Fig. R2a). We also note that both flexural branches are completely degenerate (blue markers as further highlighted by black dashed oval), which is the same as the square meta-rods.

For the rectangular meta-rods, however, the situation is more interesting. First, we see that fluid-like region is still found in z -stack meta-rod (shaded grey in Fig. R2c). However, by using a rectangular cross section, an additional anisotropy is introduced to x & y directions. We can observe that the flexural modes split into two branches (green and blue markers, also marked by black dashed rectangle) – a clear indication of the different wave speed for the two polarizations, as discussed in Fig. R1. Similar phenomenon can be seen in both x -stack cases (Fig. R2d).

We would like to further point out that longitudinal and torsional bandgaps are found in both circular and rectangular meta-rods, as shown in Fig. R2 (for longitudinal, red markers; for torsional, cyan markers in a, c, orange markers in b, d).

Here, we have only discussed the consequences of the epoxy cladding having different shapes of cross sections. If the steel core or silicone rubber coating have different cross sections, the situation will be even more complicated, and the physics become richer and more interesting. Yet to exhaustively investigate all possible combinations is far beyond the scope of this paper. Nevertheless, for all cases with changed aspect ratios in geometry, we expect that anisotropy will emerge and lead to similar phenomenon of degeneracy breaking as shown in Fig. R2.

We have added a new section in the *Supplementary Information: S-VI. Geometry of meta-rod's cross section*, to include these discussions.

The main message of our reply to this question is that all the qualitative features of polarization bandgaps are retained with a change in the cross sectional shape/aspect ratio, even though the behaviors can become more complex and rich in terms of distinct features.

Figure R2 Band structures of meta-rods with circular and rectangular cross sections. Meta-rods with circular cross section is shown in **a, b**; and meta-rods with rectangular cross section (aspect ratio 2:1) are shown in **c, d**. In all panels, red markers delineate branches associated with longitudinal modes; blue/green markers plot branches associated with flexural modes, and orange/cyan markers show torsional branches. Fluid-like elasticity can be found in both cases in z-stack meta-rods (grey shaded regions in **a, c**). The overall behaviors of the circular meta-rods are almost identical to the square meta-rods (the case studied in the main text). Rectangular meta-rods have additional anisotropy in their cross sections, consequently the two flexural branches have different propagation speed (enclosed by dashed ovals/rectangles).

What is the effect in terms of material choice and weight efficiency? i.e., how will the property change as a result of different material combinations other than the specific material pair (steel, epoxy and silicone) the authors listed?

To achieve the functionalities described in our paper, a unit cell that can function as a sub-wavelength mechanical resonator is essential. This unit cell has a very simple physical model: a mass-spring-mass harmonic resonator wherein the eigenfrequencies for modes along different spatial direction are detuned (Fig. 1a, inset). In general, this model can be translated into a three-component composite wherein a rigid and heavy core is coated by relatively soft elastic materials, and then is embedded in another relatively rigid material. And the easiest way to detune the translational eigenmodes is to make the soft layer's thickness different at different positions (The consequences of silicone layer's thickness is also numerical studied in *Supplementary Information, S-I*). Therefore we do have much freedom in the choice of materials. In this paper, we choose steel, silicone and epoxy simply because they are cheap, easily available, and easy to process in a lab. But a wide range of materials can also achieve similar or the same effects.

To demonstrate this broad freedom, we first numerically investigate how the eigenfrequency of each eigenmode changes with respect to different material parameters. The results are shown in Fig. R3. From the results, it is clear that all relevant eigenmodes can still be identified, and they can be tuned effectively with different material parameters.

Figure R3 Tuning of eigenfrequencies by material parameters. In a, mass density of the cylindrical core is altered, whereas the silicone layer and epoxy maintain their properties. It is seen

that all eigenfrequencies decrease as the core's mass density increases. In **b**, Young's modulus of the silicone is varied. Clearly, the harder (larger Young's modulus) the layer is, the higher the eigenfrequencies. Here, steel and epoxy are used as the materials of the core and cladding. In **c**, the mass density of the cladding (or background) is adjusted, whereas steel and silicone are used for the core and coating layer. Eigenfrequencies drops as the mass density increases.

As a further demonstration, we numerically studied the response functions of two new examples using some widely available materials: granite cylinder coated by foam embedded in spruce, and bronze cylinder coated by latex rubber embedded in marble. We use the following material parameters (ρ is mass density, ν is Poisson's ratio, and E is Young's modulus): $\rho = 2800 \text{ kg/m}^3$, $\nu = 0.25$, $E = 70 \text{ GPa}$ for granite; $\rho = 120 \text{ kg/m}^3$, $\nu = 0.33$, $E = 8 \text{ MPa}$ for foam; $\rho = 400 \text{ kg/m}^3$, $\nu = 0.37$, $E = 11 \text{ GPa}$ for spruce. And $\rho = 8700 \text{ kg/m}^3$, $\nu = 0.34$, $E = 120 \text{ GPa}$ for bronze; $\rho = 1300 \text{ kg/m}^3$, $\nu = 0.48$, $E = 4 \text{ MPa}$ for latex rubber; $\rho = 2700 \text{ kg/m}^3$, $\nu = 0.25$, $E = 40 \text{ GPa}$ for marble. All geometric parameters, including lattice constant, silicone thickness, cylinder core's height and diameter, etc., are kept the same. Despite some of the material properties are vastly different from the steel/silicone/epoxy combination, it is clearly observed that fluid-like elasticity and torsional bandgaps still persist, even though they occur at different frequencies as expected.

Materials: Spruce coated with Foam embedded in Granite

Materials: Bronze coated with Latex embedded in Marble

Figure S9 Polarization bandgaps realized with alternative materials. Meta-rods constructed from unit cells made of granite cylinder coated with foam embedded in spruce are shown in **a-c**. Fluid-like property can be found near 1.8–2.5 kHz (Sample- k_z , grey shaded region) and 1.8–2.4 kHz (Sample- k_x , partially fluid-like). Meta-rods constructed from unit cells made of bronze cylinder coated with latex rubber embedded in marble is shown in **d-f**. Fluid-like property can be found near 1.2–1.6 kHz (Sample- k_z , grey shaded region) and 1.2–1.7 kHz (Sample- k_x , partially fluid-like). All curves are numerically calculated using COMSOL Multiphysics.

In short, the unique phenomena demonstrated in our work is not limited to the material combination shown in main text, but can be reproduced by a variety of material choices. **These results and discussions now appear in *Supplementary Information S-VII. Material choice.***

The authors claim that these “meta-rods” can be served as an alternative method to be used for seismic shielding and demonstrated the shielding effect flexural vibrations. Can the authors comment on other types of vibrations or combinations of different vibrations?

We thank the referee for pointing out this important issue. The meta-rod’s functionality of vibration shielding roots in its polarization bandgaps. As explained in the main text, these bandgaps are the consequence of anti-crossing between local resonances and different propagation branches. Therefore the system can offer vibration shielding to different type of vibrations. We chose to emphasize the shielding effect against flexural vibrations because this type of vibration is typically responsible to more devastating consequences in seismic events.

We wish to point out that similar functionality of wave shielding can be achieved for longitudinal vibrations. This is clearly demonstrated in one of *Supplementary Videos*. We note that the frequency in the demonstration is different from that of the flexural vibration. This is because we also demonstrate fluid-like property (mismatching longitudinal and flexural bandgaps) with the same sample. Nevertheless, it is easy to shift the frequency of each polarization bandgaps by adjusting the thickness

of the silicone rubber at relevant positions (see *Supplementary Information, S-I*). **By making the unit cell isotropic, we will have a full bandgap that can effectively shield both transverse and longitudinal vibrations.** We are presenting two *Supplementary Videos* as demonstration of the vibration shielding against flexural and longitudinal vibrations, respectively. The main text is also revised to include discussions on vibration isolation against different types of vibrations.

Reviewer #3 (Remarks to the Author):

This manuscript describes the modeling and experiments associated with an “elastic metamaterial.” This consists of “meta-rods” where there are unit cells of metal cylinders encased in a polymeric matrix where there are different thicknesses of polymer on the faces of the cylinder as opposed to the curved surface. When these unit cells are arranged into rod-like structures, interesting vibrational behavior can be observed. Specifically, depending on how you orient the cylinders, the authors have been able to show band gaps in certain vibrational modes and even “fluid-like” behavior where longitudinal vibrations are transmitted while translational modes are damped. I found this paper to be interesting, novel, and very well written. The concept is well explained and the results clearly show the concept through actual testing and the discussion is easily understandable. I commend the authors on this work and a well written manuscript. This is a very interesting and powerful concept which can be applied in many ways. The authors highlight design for seismic activity at the end of the paper but there are clearly many other applications as well. This is impactful work which should be published in Nature Communications. I think that the manuscript is very complete and nearly publishable in its current state. I have the below additional comments/thoughts which are not required for.

We are excited by the referee’s high recommendation of our work, and we appreciate it very much. **We have further revised the manuscript to account for some of the referee’s suggestions.**

• One way to make this already excellent paper stronger would be if the authors could have drawn some more general conclusion. There must be some non-dimensional numbers and/or ratio of K/K' which define some of the behavior of the system. These generalizations could serve as a design guide to others.

We thank the reviewer for this suggestion. There are three material components in the unit cell, each exhibiting at least three parameters (mass density, bulk and shear moduli). Moreover, there are in fact many geometric parameters, such as the thickness of the coating silicone along different directions, the shape and volume of the core inclusion, which are both crucially important in determining the system properties. Even the cross section geometry of the meta-rod matters. Therefore, it is not very

straightforward to conclude some simple non-dimensional numbers or ratios that define the behavior of the system, for there are many degrees of freedoms. However, as demonstrated in our case, some useful conclusions can be drawn as a design guide. The metamaterial unit cell has a very simple physical model: a mass-spring-mass harmonic resonator wherein the eigenfrequencies for modes along different spatial direction are detuned (Fig. 1a, inset). In general, this can be translated into a three-component composite wherein a rigid and heavy core is coated by relatively soft elastic materials, and then is embedded in another relatively rigid material. And the easiest way to detune the translational eigenmodes is to make the soft layer's thickness different at different positions (The consequences of silicone layer's thickness is also numerical studied in *Supplementary Information, S-I*). We have also investigated the influence of cross section geometry and material choice of the system, as demonstrated in *Supplementary Information, S-VI and S-VII*.

On the other hand, we chose to use dimensional physical quantities such as Frequency (kHz) in the band structures (Fig. 2b), for the purpose of a better and more direct comparison between experimental and numerical data. We do agree that dimensionless variables do have their advantages, especially for the ease of future realization of the similar at different frequencies/scales/systems. The normalized frequency is usually defined as $\omega a/2\pi v_0$, where ω is the angular frequency, a is the lattice constant, and v_0 is the wave speed in the background medium. It is therefore straightforward to define such a normalized frequency for a system wherein v_0 is one constant. (For example, for a phononic crystal system whose background is air, speed of sound at room temperature at 1 ATM is $v_0 = 343$ m/s). However, this is not the case for elastic waves in solids: the wave speeds for each type of waves (i. e., longitudinal, flexural, torsional, or shear) are different even in a homogenous medium. Also, elastic waves can be naturally dispersive in a homogeneous medium (for example, flexural waves naturally have a quadratic dispersion relation). We therefore feel that it is a bit strange to normalize all bands and polarization branches with respect to a constant wave speed. It is therefore we choose to maintain Frequency as a dimensional quantity in the revised manuscript. On the other hand, wave vector components are already normalized (for instance, $k_z a/\pi$ as appeared in Fig. 2b).

• More thought/discussion on how to extend this concept to 3D would be very interesting.

We agree with the referee's suggestion. A thorough numerical study on a 3D lattice is presented in *Supplementary Information S-V*. We have also expanded the relevant discussions in the main text.

- **Could this concept be miniaturized and become elements within a “material”? Could the material have band gaps for specific frequencies along certain axes and provide vibrational isolation?**

Indeed, like most metamaterials, our system is scalable. Making the unit cell smaller will increase its working frequencies. Therefore, one can easily envision such structure to be working in ultrasonic frequencies. Many functionalities, for example, vibration isolation, polarization control, hyperlensing^{2,3} (via anisotropic mass density), will be possible with these units as inclusions within another host material. On the other hand, scaling up the system will lower the frequencies. With a unit cell at the size of ~2m, we found the translational modes moved to ~10Hz, which is already within seismic frequencies.

- **Finally, while mentioning other literature, the authors state on page 12 that “...existing experimental investigations focused more on their static mechanical properties, while testing of their dynamic/wave properties are lacking.” This is still generally true but starting to change. An initial foray into this area is shown in “Dynamic Behavior of Engineered Lattice Materials,” Hawreliak, et al., *Scientific Reports*, 20 June, 2016.**

We thank the reviewer for introducing this paper to us. We have read it carefully, and we agree that it should be cited. This paper now appears as Ref. 21.

References

- 1 Landau, L. & Lifshitz, E. *Theory of elasticity, course of theoretical physics*. Vol. 7 (Pergamon Press, 1986).
- 2 Lee, H., Oh, J. H., Seung, H. M., Cho, S. H. & Kim, Y. Y. Extreme stiffness hyperbolic elastic metamaterial for total transmission subwavelength imaging. *Sci. Rep.* **6** (2016).
- 3 Shen, C., Xie, Y., Sui, N., Wang, W., Cummer, S. A. & Jing, Y. Broadband acoustic hyperbolic metamaterial. *Phys. Rev. Lett.* **115**, 254301 (2015).